# Is This Loss Informative?
# Faster Text-to-Image Customization
# by Tracking Objective Dynamics

**Anton Voronov**[*]
MIPT, Yandex

**Mikhail Khoroshikh**[*]
HSE University, Yandex

**Artem Babenko**
HSE University, Yandex

**Max Ryabinin**[*]
HSE University, Yandex

## Abstract

Text-to-image generation models represent the next step of evolution in image synthesis, offering a natural way to achieve flexible yet fine-grained control over the result. One emerging area of research is the fast adaptation of large text-to-image models to smaller datasets or new visual concepts. However, many efficient methods of adaptation have a long training time, which limits their practical applications, slows down experiments, and spends excessive GPU resources. In this work, we study the training dynamics of popular text-to-image personalization methods (such as Textual Inversion or DreamBooth), aiming to speed them up. We observe that most concepts are learned at early stages and do not improve in quality later, but standard training convergence metrics fail to indicate that. Instead, we propose a simple drop-in early stopping criterion that only requires computing the regular training objective on a fixed set of inputs for all training iterations. Our experiments on Stable Diffusion for 48 different concepts and three personalization methods demonstrate the competitive performance of our approach, which makes adaptation up to 8 times faster with no significant drops in quality.

## 1 Introduction

Large text-to-image synthesis models have recently attracted the attention of the research community due to their ability to generate high-quality and diverse images that correspond to the user's prompt in natural language [1, 2, 3, 4, 5]. The success of these models has driven the development of new problem settings that leverage their ability to draw objects in novel environments. One particularly interesting task is *personalization* (or *adaptation*) of text-to-image models to a small dataset of images provided by the user. The goal of this task is to learn the precise details of a specific object or visual style captured in these images: after personalization, the model should be able to generate novel renditions of this object in different contexts or imitate the style that was provided as an input.

Several recent methods, such as Textual Inversion [6], DreamBooth [7], and Custom Diffusion [8], offer easy and parameter-efficient personalization of text-to-image models. Still, a major obstacle on the path to broader use of such methods is their computational inefficiency. Ideally, models should adapt to user's images in real or close to real time. However, as noted by the authors of the first two papers, the training time of these methods can be prohibitively long, taking up to two hours for a single concept. The reported training time for Custom Diffusion is 12 minutes per concept on a single GPU, which is much faster but still outside the limits of many practical applications.

---

[*]Equal contribution. Correspondence to `mryabinin0@gmail.com`

37th Conference on Neural Information Processing Systems (NeurIPS 2023).

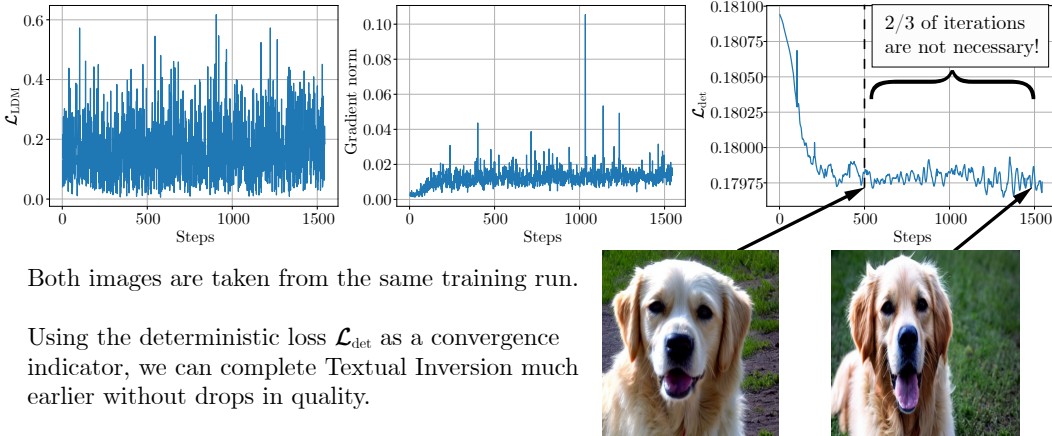

Both images are taken from the same training run.

Using the deterministic loss $\mathcal{L}_{\text{det}}$ as a convergence indicator, we can complete Textual Inversion much earlier without drops in quality.

Figure 1: A summary of our key findings: the quality of methods like Textual Inversion saturates early on, but the training loss does not indicate that. Evaluating the same loss on a batch of data fixed throughout the run makes the training dynamics significantly more informative.

In this work, we seek answers to the following questions: **are text-to-image customization techniques inherently time-consuming** and **can we decrease their runtime without major quality tradeoffs?** Focusing on the training dynamics, we observe that the CLIP [9] image similarity score (often used to assess image quality in such tasks [10]) grows sharply only in the early stages of Textual Inversion and hardly improves after that. However, as shown in Figure 1, neither the training loss nor the gradient norm indicate the convergence of the concept embedding, which prevents us from stopping the adaptation process earlier. While it is possible to score samples from the model with CLIP during training, generating them can take quite a long time.

With that in mind, we study the training objective itself and attempt to understand the reasons of its non-informative dynamics. As we demonstrate, the primary cause lies in several sources of stochasticity (such as diffusion time steps or the diffusion noise) introducing noise to the loss function. If we sample random variables only once and fix them across training steps, the loss for these inputs reflects convergence better even when training with the *original* (fully stochastic) objective.

Motivated by this finding, we propose **D**eterministic **VAR**iance Evaluation (DVAR), an early stopping criterion for text-to-image customization methods that generates a single fixed batch of inputs at the beginning of training and evaluates the model on this batch after each optimization step. This criterion is straightforward to implement, has two interpretable hyperparameters, and its stopping time corresponds to convergence in terms of the CLIP image score. We validate DVAR by comparing it with a range of baselines on three popular adaptation methods, showing that it is possible to run methods like Textual Inversion and DreamBooth much faster with up to 8x speedups and little to no decline in quality. For Custom Diffusion, our method recovers the optimal number of training iterations, which makes it useful to avoid empirical tuning of the step count for every specific dataset.

In summary, our contributions are as follows:

- We investigate the difficulties in detecting the convergence of text-to-image adaptation methods from training-time metrics. As we demonstrate, the objective becomes much more interpretable across iterations if computed on the same batch of inputs without resampling any random variables.

- We propose DVAR, a simple early stopping criterion for text-to-image personalization techniques. This criterion is easy to compute and use, does not affect the training process and correlates with convergence in terms of visual fidelity.

- We compare this criterion with multiple baselines on Stable Diffusion v1.5, a popular text-to-image model, across three methods: Textual Inversion, DreamBooth, and Custom Diffusion.[1] DVAR offers a significant decrease in runtime while having quality comparable to both original methods and other early stopping baselines.

---

[1]The code of our experiments is available at `github.com/yandex-research/DVAR`.

## 2  Background

### 2.1  Denoising Diffusion Probabilistic Models

Diffusion models [11] are a class of generative models that has become popular in recent years due to their ability to generate both diverse and high-fidelity samples [12]. They approximate the data distribution through iterative denoising of a variable $z_t$ sampled from Gaussian distribution. In simple words, the model $\epsilon_\theta$ is trained to predict the noise $\epsilon$, following the objective below:

$$\min_\theta \mathbb{E}_{z_0, \, \epsilon \sim N(0,I), \, t \sim U[1,T]} ||\epsilon - \epsilon_\theta(z_t, c, t)||_2^2. \tag{1}$$

Here, $z_t$ corresponds to a Markov chain *forward* process $z_t(z_0, \epsilon) = \sqrt{\alpha_t} z_0 + \sqrt{1 - \alpha_t} \epsilon$ that starts from a sample $z_0$ of the data distribution. For example, in the image domain, $z_0$ corresponds to the target image, and $z_T$ is its version with the highest degree of corruption. In general, $c$ can represent the condition embedding from any other modality. The inference (*reverse*) process occurs with a fixed time step $t$ and starts from $z_t$ equal to a sample from the Gaussian distribution.

### 2.2  Text-to-image generation

The most natural and well-studied source of guidance for generative models is the natural language [13, 14, 15] because of its convenience for the user, ease of collecting training data, and significant improvements in text representations over the past several years [5]. The condition vector $c$ is often obtained from Transformer [16] models like BERT [17] applied to the input text.

State-of-the-art text-to-image models perform forward and reverse processes in the latent space of an autoencoder model $z_0 = \mathcal{E}(x), x = \mathcal{D}(z_0)$, where $x$ is the original image, and $\mathcal{E}, \mathcal{D}$ are the encoder and the decoder, respectively. We experiment with Stable Diffusion v1.5 [3], which uses a variational autoencoder [18] for latent representations. Importantly, this class of autoencoder models is not deterministic, which makes inference even more stochastic but leads to a higher diversity of samples. In total, given image and caption distributions $X, Y$, the training loss can be formulated as:

$$\mathcal{L}_{train} = \mathbb{E}_{y,x,\epsilon} ||\epsilon - \epsilon_\theta(z_t(\mathcal{E}(x), \epsilon), c(y), t)||_2^2, \tag{2}$$

$$y \sim Y, x \sim X, \epsilon \sim \mathcal{N}(0, I), t \sim U[0, T] \tag{3}$$

### 2.3  Adaptation of text-to-image models

While text-to-image generation models are flexible because of the natural language input, it is often difficult to design a prompt that corresponds to an exact depiction of an object of choice. Hence, several methods for adapting such models to a given collection of images were developed. In this work, we focus on the most well-known methods, discussing others in Appendix A.

Most approaches inject the target concept into the text-to-image model by learning the representation of a new token $\hat{v}$ inserted into the language encoder vocabulary. The representation is learned by optimizing the reconstruction loss from Equation 2 for a few (typically $3 - 5$) reference images $I$ with respect to the embedding of $\hat{v}$, while other parts of the model are frozen. The main advantage of these methods is the ability to flexibly operate with the "pseudo-word" $\hat{v}$ in natural language, for example, by placing the concept corresponding to that word into different visual environments.

Textual Inversion [6] is the simplest method for adaptation of text-to-image models that updates only the token embedding for $\hat{v}$. While this method is parameter-efficient, its authors report that reaching acceptable inversion quality requires 6000 training iterations, which equals $\approx$2 GPU hours on most machines. This number is a conservative estimate of the number of steps sufficient for all concepts: authors of the method propose selecting the earliest checkpoint that exhibits sufficient train image reconstruction quality[2].

While Textual Inversion only learns the embedding of the target token, DreamBooth [7] does the same with a fully unfrozen U-Net component, and Custom Diffusion [8] updates the projection matrices of cross-attention layers that correspond to keys and values. These methods use a significantly smaller number of iterations, 1000 and 500 respectively, which is also not adaptive and can lead to overfitting.

---

[2]`github.com/rinongal/textual_inversion/issues/34#issuecomment-1230831673`

# 3 Understanding adaptation dynamics

As we explained previously, the goal of our study is to find ways of speeding up text-to-image without significantly degrading the quality of learned concept representations. To accomplish this goal, we analyze the optimization process of running Textual Inversion on a given dataset. As we demonstrate in Section 4 and in Appendix B, the same findings hold for DreamBooth and Custom Diffusion, which makes our analysis broadly applicable to different methods for customized text-to-image generation.

We apply Textual Inversion to concepts released by Gal et al. [6], using Stable Diffusion v1.5[3] as the base model. We monitor several metrics during training:

1. First, one would hope to observe that optimizing the actual training objective would lead to its convergence, and thus we track the value of $\mathcal{L}_{train}$.

2. Second, we monitor the gradient norm, which is often used for analyzing convergence in non-convex optimization. As the model converges to a local optimum, the norm of its gradient should also decrease to zero or stabilize at a point determined by the gradient noise.

3. Lastly, every 50 training iterations, we generate 8 samples from the model using the current concept embedding and score them with the CLIP image similarity score using the training set as references. In the original Textual Inversion paper, this metric is named the reconstruction score and is used for quantitative evaluation.

Note that we do not rely on the CLIP text-image score for captions: in our preliminary experiments, we observed no identifiable dynamics for this metric when using the entire set of CLIP caption templates. Writing more specific captions and choosing the most appropriate ones for each concept takes substantial manual effort; hence, we leave caption alignment out of the scope for this study.

## 3.1 Initial observations

First, we would like to view the training dynamics in terms of extrinsic evaluation: by measuring how the CLIP image score changes throughout training, we can at least estimate how fast the samples begin to resemble the training set. For this, we perform inversion of all concepts released by [6, 7, 8] (a total of 48 concepts): an example of such an experiment for one concept is available in Figure 2.

From these experiments, we observe that the CLIP image score exhibits sharper growth at an early point of training (often within the first 1000 iterations) and stabilizes later. This finding agrees with the results of our own visual inspection: the generated samples for most concepts undergo **the most drastic changes at the beginning** and do not improve afterwards. Practically, this observation means that we can interrupt the inversion process much earlier without major drawbacks if we had a criterion for detecting its convergence. What indicators can we use to create such a criterion?

The most straightforward idea is to consider the training loss $\mathcal{L}_{train}$. Unfortunately, it is not informative by default: as we demonstrate in Figure 2, the loss exhibits too much noise and has no signs of convergence. The gradient norm of the concept embedding is also hardly informative: in the same experiment, we can see that it actually increases during training instead of decreasing. As we show in Appendix C, these findings generally hold even for much larger training batches, which means that the direct way of making the loss less stochastic is not practical. Still, as reported in the original work and shown by samples and their CLIP scores, the model successfully learns the input concepts. Curiously, we see no reflection of that in the dynamics of the training objective.

Another approach to early stopping is to leverage our observations about the CLIP image score and measure it during training, terminating when the score fails to improve for a specific number of iterations. However, there are two downsides to this approach. First, frequently sampling images during training significantly increases the runtime of the method. Second, this criterion can be viewed as directly maximizing the CLIP score, which is known to produce adversarial examples for CLIP instead of actually improving the image quality [2].

---

[3]We chose this version because it was given as a default in the Diffusers [19] example for Textual Inversion.

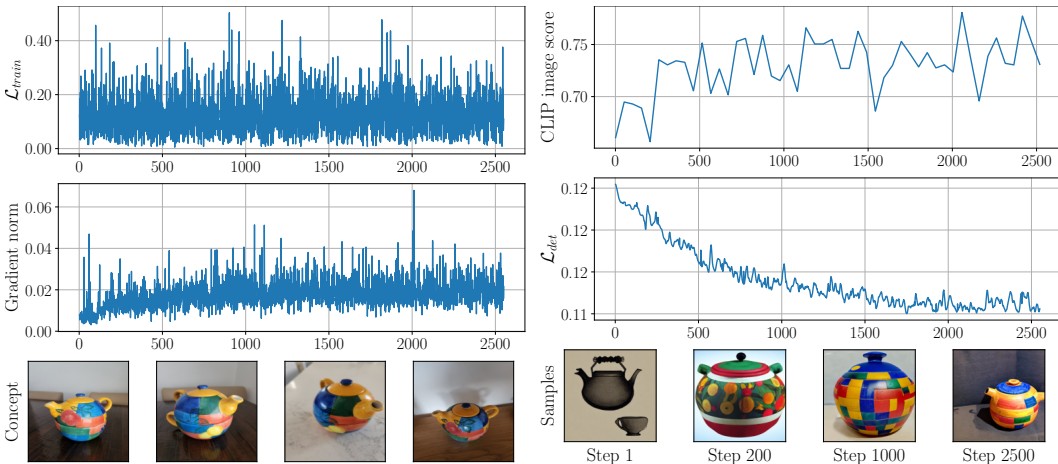

Figure 2: An overview of the convergence process for Textual Inversion with an example concept. Figures 6 and 7 in Appendix B present more examples.

## 3.2 Investigating the sources of randomness

We hypothesize that the cause of excessive noise in $\mathcal{L}_{train}$ is several factors of randomness injected at each training step, as we mentioned previously in Section 2.2. Thus, we aim to estimate the influence of the following factors on the dynamics of the inversion objective:

1. Input images $x$
2. Captions corresponding to images $y$
3. VAE Latent representations for images $\mathcal{E}(x)$
4. Diffusion time steps $t$
5. Gaussian diffusion noise $\epsilon$

Now, our goal is to identify the factors of stochasticity that affect the training loss. Importantly, we **do not change** the **training** batches, as it alters the objective of Textual Inversion and might affect its outcome. Thus, we train the model in the original setting (with batches of entirely random data) but evaluate it on batches with some sources of randomness **fixed** across all iterations. Note that $\mathcal{E}(x)$ depends on $x$: if we resample the input images, we also need to resample their latent representations.

First, we try the most direct approach of making *everything* deterministic: in other words, we compute $\mathcal{L}_{det}$, which is the same as $\mathcal{L}_{train}$, but instead of the expectation over random data and noise, we compute it on the same inputs after each training step. Formally, we can define it as

$$\mathcal{L}_{det} = ||\epsilon - \epsilon_\theta(z_t(\mathcal{E}(x), \epsilon), c(y), t)||_2^2, \tag{4}$$

with $x$, $y$, $\mathcal{E}(x)$, $t$, and $\epsilon$ sampled only once in the beginning of inversion. Essentially, this means that the only argument of this function that changes across training iterations is $c(y)$ that depends on the trained concept embedding.

As we show in Figure 2, this version of the objective becomes informative, indicating convergence across a broad range of concepts. Moreover, it displays approximately the same behavior as the CLIP score and is much less expensive to compute, which makes $\mathcal{L}_{det}$ particularly useful as a metric for the stopping criterion.

For the next step, we aim to find if any of the above sources of stochasticity have negligible impact on the noise in $\mathcal{L}_{train}$ or can be compensated with larger batches. We evaluate them separately and provide results in Section 4.5. Our key findings are that (1) resampling **captions and VAE encoder noise** still **preserves the convergence trend**, (2) using **random images or resampling diffusion noise** reveals the training dynamics **only for large batches**, and (3) sampling **different diffusion timesteps** leads to a **non-informative training loss** regardless of the batch size. Still, for the sake of simplicity and efficiency, we compute $\mathcal{L}_{det}$ on a batch of 8 inputs sampled only once at the beginning of training for the rest of our experiments.

```
def DVAR(losses, window_size, threshold):
    running_var = losses[-window_size:].var()
    total_var = losses.var()
    ratio = running_var / total_var
    return ratio < threshold
```

Figure 3: An example NumPy/PyTorch implementation of DVAR. See Appendix D for an example of its usage in the training code.

### 3.3 Deterministic Variance Evaluation

The results above show that fixing all random components of the textual inversion loss makes its dynamics more interpretable. To achieve our final goal of decreasing the inversion runtime, we need to design an early stopping criterion that leverages $\mathcal{L}_{det}$ to indicate convergence.

We propose Deterministic Variance Evaluation (DVAR), a simple variance-based early stopping criterion. It maintains a rolling variance estimate of $\mathcal{L}_{det}$ over the last $N$ steps, and once this rolling variance becomes less than $\alpha$ of the global variance estimate ($0 < \alpha < 1$), we stop training. A pseudocode implementation of DVAR is shown in Figure 3.

This criterion is easy to implement and has two hyperparameters that are easy to tune: the window size for local variance estimation $N$ and the threshold $\alpha$. In our experiments, we found $\{N = 310, \alpha = 0.15\}$ for Textual Inversion, $\{N = 440, \alpha = 0.4\}$ for DreamBooth, and $\{N = 180, \alpha = 0.15\}$ for Custom Diffusion to work relatively well across all concepts we evaluated.

Importantly, we use this criterion while training in the **regular fully stochastic setting**: our goal is not to modify the objective, and using fixed random variables and data can affect the model's generalization capabilities. As we demonstrate in Section 4, our approach demonstrates significant improvements when compared to baselines, even when all sources of randomness are fixed.

Along with DVAR, we tested other early stopping strategies that use $\mathcal{L}_{det}$ and are based on different notions of loss value stabilization, such as estimating the trend coefficient or tracking changes in the mean instead of variance. As we show in Appendix E, most of them result in less reliable convergence indicators and have hyperparameters that do not transfer as well between different image collections.

## 4 Experiments

In this section, we compare DVAR with several baselines in terms of sample fidelity for the learned concept, output alignment with novel prompts, and training time. Our goal is to verify that this early stopping criterion is broadly applicable and has little impact on the outcome of training.

### 4.1 Setup and data

We evaluate approaches to early stopping on three popular text-to-image personalization methods: Textual Inversion, DreamBooth with LoRA [20], and Custom Diffusion, all applied to Stable Diffusion v1.5[4]. For Textual Inversion, we use the implementation from the Diffusers library[5] [19] and take the hyperparameters from the official repository[6]. We use the official code and hyperparameters[7] for Custom Diffusion, and we use the implementation of DreamBooth-LoRA from the Diffusers library[8].

For evaluation, we combine the datasets published by authors of the three techniques above that were available as of March 2023, which results in a total of 48 concepts. Our main results are obtained by applying each method to all of these concepts, which in total took around 80 GPU hours. Each experiment used a single NVIDIA A100 80GB GPU. Although the methods we study can be used with several concepts at a time, we train on the images of each concept separately. Following Equation 2, the training batches are generated from a small set of original images with randomly sampled augmentations (central/horizontal crop), captions, timesteps and diffusion noise.

---

[4]huggingface.co/runwayml/stable-diffusion-v1-5
[5]github.com/huggingface/diffusers/tree/main/examples/textual_inversion
[6]github.com/rinongal/textual_inversion
[7]github.com/adobe-research/custom-diffusion
[8]github.com/huggingface/diffusers/tree/main/examples/dreambooth-lora

To automatically assess the quality of generated images, we employ CLIP image-to-image similarity for images generated from prompts used during training ("Train CLIP img"), which allows us to monitor the degree to which the target concept is learned. However, relying solely on this metric can be risky, as it does not capture overfitting. To measure generalization, we utilize CLIP text-to-image similarity, evaluating the model's ability to generate the concept in new contexts with prompts that are not seen at training time ("Val CLIP txt"). For each method, we report both the number of iterations and the average adaptation runtime, since the average duration of one iteration may differ due to intermediate image sampling or additional loss computation. In addition, we evaluate the identity preservation and diversity of all methods in Appendix F.

## 4.2 Baselines

We compare DVAR with several baselines: the original training setup of each adaptation method (named as "Baseline"), early stopping based on the CLIP similarity of intermediate samples to the training set ("CLIP-s"), as well as the original setup with the reduced number of iterations and no intermediate sampling ("Few iters").

Specifically, the original setup runs for a predetermined number of steps (6100, 1000, and 500 for Textual Inversion, Dreambooth-LoRA, and Custom Diffusion accordingly), sampling 8 images every 500/100/100 iterations and computing their CLIP similarity to the training set images. For each customization method, we tune the hyperparameters of CLIP-s: sampling frequency (number of training iterations between generating samples), as well as the threshold for minimal improvement and the number of consecutive measurements without improvement before stopping. These hyperparameters are chosen on a held-out set of 4 concepts to achieve 90% of the maximum possible train CLIP image score for the least number of iterations. The final checkpoint is selected from the iteration with the best train CLIP image score.

After running CLIP-s on all concepts, we determine the average and maximum number of steps before early stopping across concepts. Then, we run the baseline method with a reduced number of iterations and no intermediate sampling. This baseline can serve as an estimate of the minimum time required for the adaptation to converge, as it has the most efficient iterations among all methods. However, in real-world scenarios and for new customization approaches, the exact number of iterations is unknown in advance, which makes this method less applicable in practice, as it would involve rerunning such large-scale experiments for every new approach.

## 4.3 Results

The results of our experiments are shown in Table 1. As we can see, DVAR is more efficient than the baseline and CLIP-s in terms of the number of iterations and overall runtime, approaching the performance of "Few iters" (which can be considered an oracle method) while being fully input-adaptive and not relying on costly intermediate sampling. In particular, the fixed step count of "Few Iters" results in higher standard deviations for almost all metrics: from a practical standpoint, it means that this number of iterations might be either sufficient or excessive when adapting a model to a given set of images. We additionally discuss the need for the adaptive number of iterations in Appendix G.

Furthermore, although CLIP-s and the original setup optimize the train CLIP image score by design (in case of the baseline, this metric is used for choosing the best final checkpoint), DVAR is able to achieve nearly the same final results. Finaly, with a sufficiently long training time, the increase in the train CLIP image score leads to a decline in the validation CLIP text score, which indicates overfitting on input images: this phenomenon is illustrated in more detail in Figure 4 and Appendix H. As we can see from the CLIP text scores of DVAR, early stopping helps mitigate this issue.

## 4.4 Human evaluation

To verify the validity of our findings, we compare the quality of images obtained with DVAR with that of the original methods. To do that, we used a crowdsourcing platform, asking its users to compare samples of the baseline approach and samples from the model finetuned for the number of iterations determined by DVAR.

We conducted two surveys: the first survey compared samples derived from prompts seen by the model during training, measuring the reconstruction ability of adaptation. Participants were instructed

Table 1: Comparison of speedup methods for three approaches to text-to-image personalization. The best value is in bold.

| Method | Train CLIP img | Val CLIP img | Val CLIP txt | Iterations | Time, min |
|---|---|---|---|---|---|
| **Textual Inversion** | | | | | |
| Baseline | $0.840_{\pm 0.051}$ | $0.786_{\pm 0.075}$ | $0.209_{\pm 0.021}$ | 6100 | $27.0_{\pm 0.3}$ |
| CLIP-s | $0.824_{\pm 0.053}$ | $0.757_{\pm 0.067}$ | $0.233_{\pm 0.024}$ | $666.7_{\pm 174.5}$ | $9.6_{\pm 2.5}$ |
| Few iters (mean) | $0.796_{\pm 0.069}$ | $0.744_{\pm 0.073}$ | $0.232_{\pm 0.023}$ | **475** | $\mathbf{1.6_{\pm 0.0}}$ |
| Few iters (max) | $0.806_{\pm 0.066}$ | $0.767_{\pm 0.071}$ | $0.219_{\pm 0.022}$ | 850 | $2.8_{\pm 0.0}$ |
| DVAR | $0.795_{\pm 0.067}$ | $0.748_{\pm 0.068}$ | $0.227_{\pm 0.024}$ | $566.0_{\pm 141.5}$ | $3.1_{\pm 0.8}$ |
| **DreamBooth-LoRA** | | | | | |
| Baseline | $0.865_{\pm 0.045}$ | $0.833_{\pm 0.067}$ | $0.203_{\pm 0.019}$ | 1000 | $7.8_{\pm 1.5}$ |
| CLIP-s | $0.862_{\pm 0.045}$ | $0.788_{\pm 0.075}$ | $0.225_{\pm 0.022}$ | $353.2_{\pm 88.1}$ | $6.1_{\pm 1.5}$ |
| Few iters (mean) | $0.855_{\pm 0.052}$ | $0.806_{\pm 0.085}$ | $0.219_{\pm 0.023}$ | **367** | $\mathbf{2.0_{\pm 0.1}}$ |
| Few iters (max) | $0.851_{\pm 0.053}$ | $0.800_{\pm 0.097}$ | $0.214_{\pm 0.019}$ | 500 | $2.6_{\pm 0.1}$ |
| DVAR | $0.784_{\pm 0.106}$ | $0.687_{\pm 0.140}$ | $0.238_{\pm 0.034}$ | $665.3_{\pm 94.9}$ | $4.9_{\pm 0.7}$ |
| **Custom Diffusion** | | | | | |
| Baseline | $0.755_{\pm 0.077}$ | $0.695_{\pm 0.069}$ | $0.258_{\pm 0.021}$ | 500 | $6.5_{\pm 0.9}$ |
| CLIP-s | $0.757_{\pm 0.076}$ | $0.691_{\pm 0.069}$ | $0.258_{\pm 0.023}$ | $510.4_{\pm 134.1}$ | $9.7_{\pm 3.0}$ |
| Few iters (mean) | $0.751_{\pm 0.078}$ | $0.691_{\pm 0.070}$ | $0.259_{\pm 0.023}$ | $450.0_{\pm 0.0}$ | $\mathbf{3.4_{\pm 0.9}}$ |
| Few iters (max) | $0.754_{\pm 0.078}$ | $0.691_{\pm 0.073}$ | $0.257_{\pm 0.022}$ | $700.0_{\pm 0.0}$ | $5.3_{\pm 1.4}$ |
| DVAR | $0.742_{\pm 0.074}$ | $0.693_{\pm 0.066}$ | $0.259_{\pm 0.022}$ | $\mathbf{348.1_{\pm 46.6}}$ | $\mathbf{3.4_{\pm 1.0}}$ |

to select the image that resembled the reference object the most. The second survey compared samples generated using novel prompts, measuring the final model capability for customization. Participants were asked to determine which image corresponded better to the given prompt. Each pair of images was assessed by multiple participants, ensuring an overlap of 10 responses. For each set of 4 responses, we paid 0.1\$, and the annotation instructions can be viewed in Appendix I.

Our findings obtained from these evaluations are shown in Table 2. As we can see, DVAR allows for early stopping without sacrificing the reconstruction quality for two out of three evaluated customization methods. Hence, our approach enables efficient training and reduces the computational cost of finetuning on new concepts. While applying DVAR to Textual Inversion slightly reduces the reconstruction quality, this can be attributed to the overfitting of the original method: other methods, which use significantly fewer iterations, are able to avoid this issue.

Table 2: Human evaluation results: the percentage of cases where the quality of DVAR was better or equal to Baseline quality.

| Method | Reconstruction | Customization |
|---|---|---|
| Textual Inversion | 41.6 | 77.7 |
| DreamBooth-LoRA | 67.9 | 91.4 |
| Custom Diffusion | 69.9 | 93.8 |

## 4.5 Analysis and ablation

Having demonstrated the advantages of DVAR, we now perform a more detailed analysis of the effect of the changes we introduce to the procedure of text-to-image adaptation. We aim to answer a series of research questions around our proposed criterion and the behavior of the training objective with all factors of variation fixed across iterations.

**Is it possible to observe convergence without determinism?** To answer this question, we conduct a series of experiments where we "unfix" each component responsible for the stochasticity of the training procedure one by one. For each component, we aim to find the smallest size of the batch that preserves the indicativity of the evaluation loss. For large batch sizes, we accumulate losses from several forward passes.

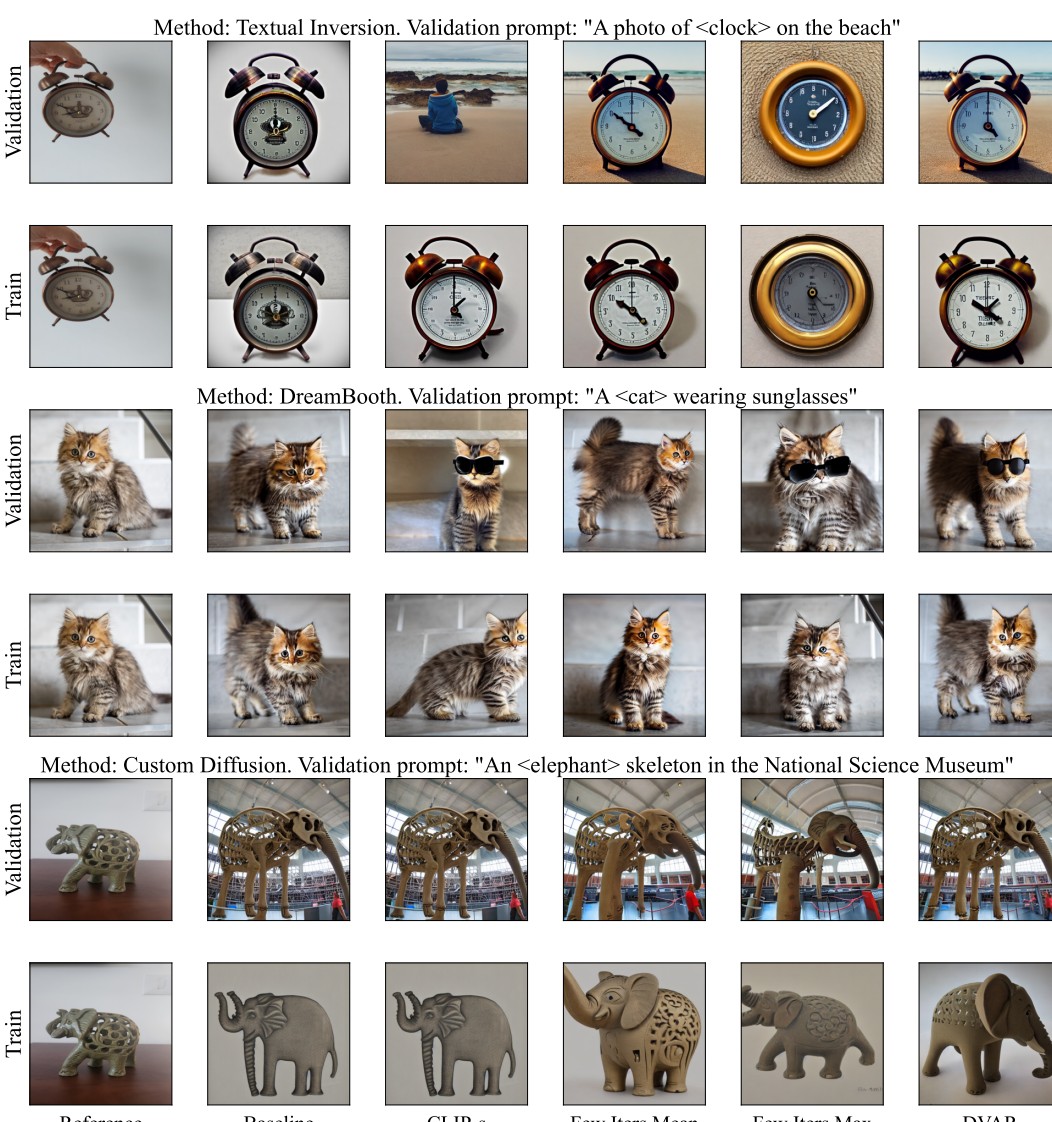

Figure 4: Comparison of early stopping techniques applied to personalization methods.

Figure 5 shows the results of these experiments. While for some parameters increasing the batch size leads to a more evident trend in loss, the stochasticity brought by varying the timesteps $t$ on each iteration cannot be eliminated even with batches as large as 512. This can be explained by the scale of the reconstruction loss being highly dependent on the timestep of the reverse diffusion process. Thus, aggregating it over multiple different points makes it completely uninformative, as the scale of the trend is several orders of magnitude less than the average values of the loss.

**Is it possible to use smaller batches for evaluation and still detect convergence?** To address this question, we ran experiments with deterministic batches of size $1, 2$ and $4$ for multiple concepts and generally observed the same behavior (as illustrated in Figure 8 of Appendix C). However, the dynamics of the objective become more dependent on the timesteps that are sampled: as a result, the early stopping criterion becomes more brittle. Hence, we use a batch size of 8 in our main experiments, as it corresponds to a reasonable tradeoff between stability and computational cost.

**Does the selection of diffusion timesteps influence the behavior of the deterministic objective?** In our approach, timesteps are uniformly sampled from a range of 0 to 1000 once at the beginning of the training process. However, according to the analysis presented in prior work [15], timesteps from

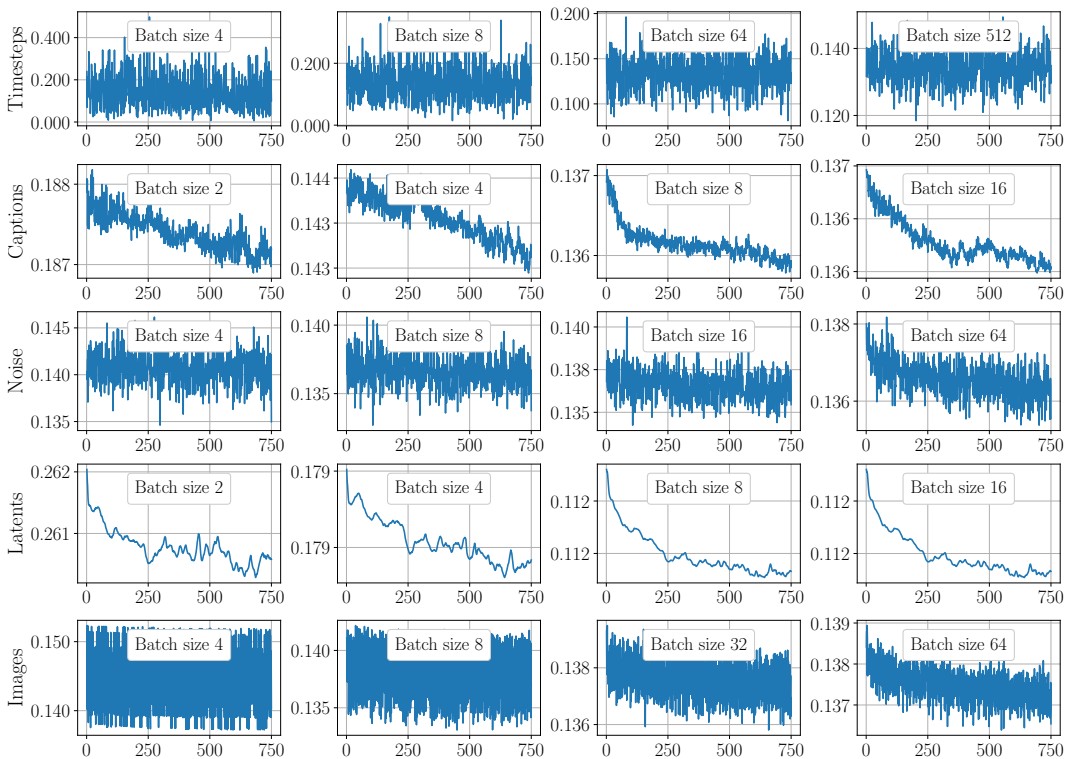

Figure 5: Loss behavior in the semi-deterministic setup: row names correspond to inputs that are resampled for each evaluation batch.

different ranges have varying effects on the results of text-to-image generation. This raises a question of whether it is possible to sample timesteps only from a specific range, as it might lead to a higher correlation between the loss and the target metrics. Our findings in Appendix J indicate that there is no significant difference in correlations when we change the sampling interval.

## 5  Conclusion

In this paper, we analyze the training process of text-to-image adaptation and the impact of different sources of stochasticity on the dynamics of its objective. We show that removing all these origins of noise makes the training loss much more informative during training. This finding motivates the creation of DVAR, a simple early stopping criterion that monitors the stabilization of the deterministic loss based on its running variance. Through extensive experiments, we verify that DVAR reduces the training time by 2 – 8 times while still achieving image quality similar to baselines.

Our work offers a robust and efficient approach for monitoring the training dynamics without requiring computationally expensive procedures. Moreover, it has adaptive runtime depending on the input concept, which naturally mitigates overfitting. These advantages make DVAR a useful tool for text-to-image researchers and practitioners, as it provides a more efficient and controlled training process, ultimately leading to improved results while avoiding the unnecessary computational overhead.

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

# A  Additional related work

Most existing methods for text-to-image personalization achieve the goal by finetuning diffusion models. In addition to the approaches evaluated in our work, there exist many other techniques that were recently proposed. For example, in $P+$ [21], the concept is inverted into a sequence of per-layer tokens, and SVDiff [22] focuses on learning the singular values of the SVD decomposition of weight matrices. Imagic [23] divides the adaptation process into two phases. In the first phase, this method optimizes the embedding in the space of texts instead of tokens, and in the second one, it trains the model separately to reconstruct the target image from the embedding. The optimized and target embeddings can then be used for image manipulation by interpolating between them. Furthermore, adaptation can also be performed with gradient-free methods using evolutionary algorithms [24].

Text-to-image model customization can also be broadly viewed as an instance of image editing: given the caption and the image generated from it, we want to change the picture according to the new prompt. There are several methods in this field that use diffusion models and have similar problem statements. For example, Prompt-to-Prompt [25] solves the task by injecting the new prompt into cross-attention, and MagicMix [26] replaces the guidance vector after several steps during inference. The intrinsic disadvantage of such methods is the need for an additional inversion step for editing existing images, like DDIM [27] or Null-text Inversion [28].

# B  Indicators of convergence for other customization methods

We provide examples of the behavior of standard model convergence indicators for other techniques for text-to-image personalization in Figures 6 and 7. While the reconstruction loss and the gradient norm are still hardly informative, both the CLIP image score and $\mathcal{L}_{det}$ exhibit a trend that corresponds to improvements in the visual quality of samples.

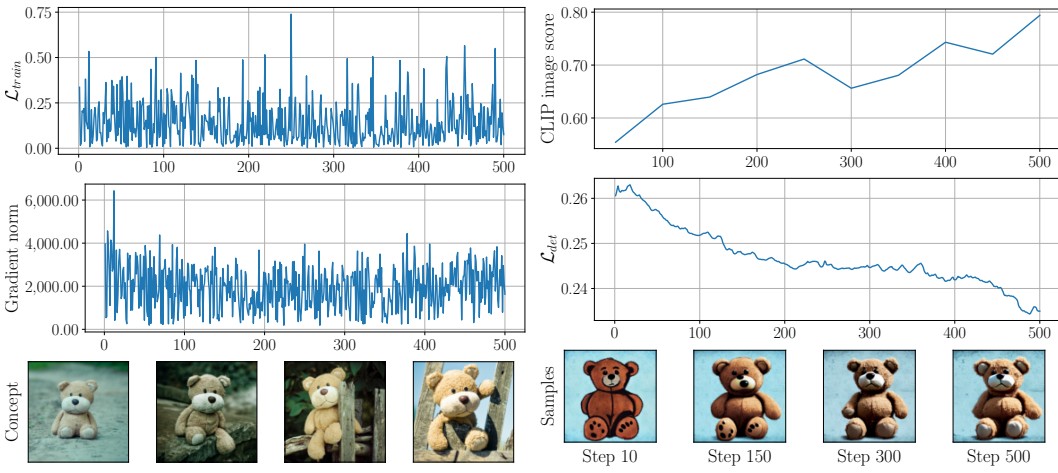

Figure 6: Convergence process for Custom Diffusion applied to an example concept.

# C  The impact of the batch size on convergence dynamics

As we increase the batch size for our evaluation objective, we sample a more diverse range of timesteps, making the loss less dependent on the random seed and more indicative of convergence. Figure 8 provides an illustration of this phenomenon.

Moreover, we provide the results of training in a regular Textual Inversion setup with much larger batches. As we demonstrate in Figure 9a, the loss dynamics remain the same even if we increase the batch size to 128, which is 64 times larger than the default one and thus requires 64x more time for a single optimization step.

Although the gradient norm of training runs with extremely large batches begins to display convergence (see Figure 9b), using the batches of this size is less practical than using an early stopping

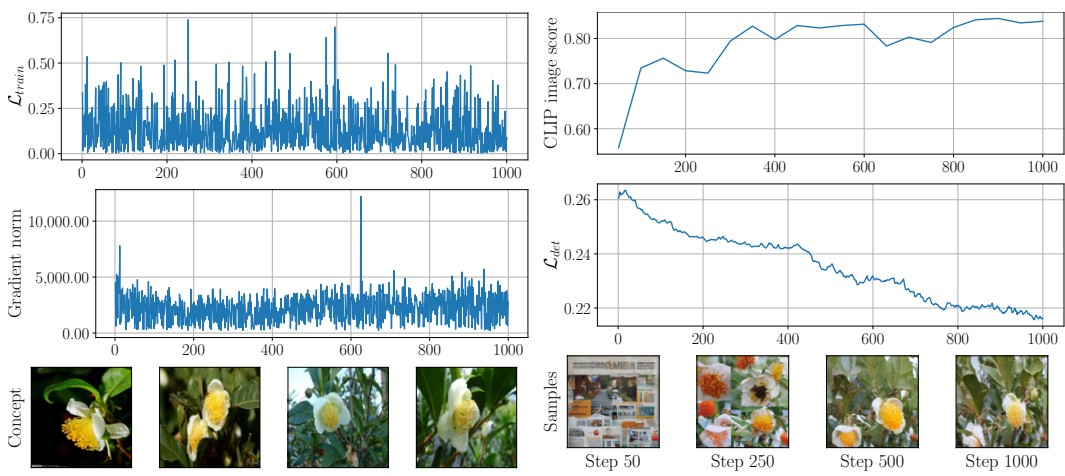

Figure 7: Convergence process for DreamBooth applied to an example concept.

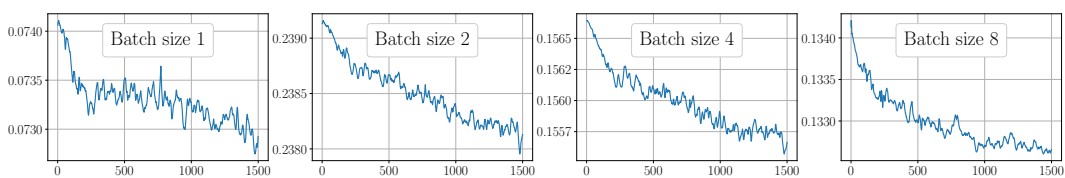

Figure 8: $\mathcal{L}_{det}$ behavior in the fully deterministic setup with varying batch sizes.

criterion. Specifically, when training with these batch sizes, the quality of samples reaches its peak at approximately 250 training steps. This corresponds to $\approx 16{,}000$ forward and backward passes with the batch size of 4 (or 8,000 with the batch size of 8 — the largest size that fits into an 80GB GPU for Stable Diffusion) and thus gives no meaningful speedup to the inversion procedure.

# D   Example pseudocode of using DVAR for training

In Figure 10, we provide an example PyTorch training loop that uses an implementation of DVAR defined in Figure 3. This code samples the evaluation batch once before training and computes $\mathcal{L}_{det}$ on it after each training iteration. A full implementation of training with DVAR is given at `github.com/yandex-research/DVAR`.

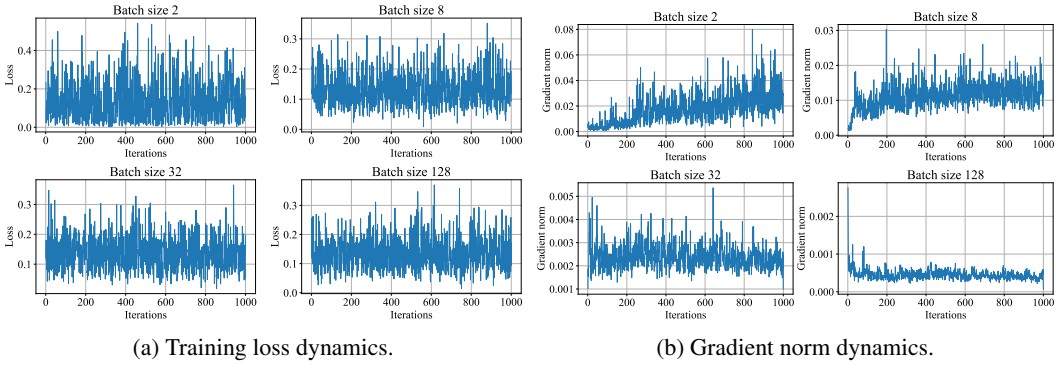

(a) Training loss dynamics.              (b) Gradient norm dynamics.

Figure 9: Training dynamics of Textual Inversion for a single concept with different batch sizes.

```python
def training_loop(model, train_dataloader, eval_losses, eval_batch,
                  optimizer, window_size, threshold):

    eval_images, eval_captions = eval_batch
    eval_stochastic = sample_everything(eval_batch)
    eval_latents, eval_timesteps, eval_noisy_latents = eval_stochastic

    for train_batch in train_dataloader:
        images, captions = train_batch
        latents, timesteps, noisy_latents = sample_everything(train_batch)

        optimizer.zero_grad()
        denoised_latents = model(captions, noisy_latents, timesteps)
        train_loss = F.mse_loss(denoised_latents, latents)
        train_loss.backward()
        optimizer.step()

        with torch.no_grad():
            eval_denoised_latents = model(
                eval_captions,
                eval_noisy_latents,
                eval_timesteps
            )
            eval_loss = F.mse_loss(eval_denoised_latents, eval_latents)
            eval_losses.append(eval_loss)

        if DVAR(torch.stack(eval_losses), window_size, threshold):
            break
```

Figure 10: PyTorch-like pseudocode of training with DVAR.

# E   Evaluation of other stopping criteria

Besides the ratio of the rolling variance to the global variance used in DVAR, we also experimented with other metrics for early stopping that consider the behavior of $\mathcal{L}_{det}$. All of these metrics have a hyperparameter $n$ that denotes the number of last objective values to use for computing aggregates.

To obtain the *EMA percentile*, we calculate the exponential moving average (with $\alpha = 0.1$) at the moment $t$ and $n$ steps back, then apply the following formula:

$$\mathcal{M}_{EMA} = \frac{EMA(t) - EMA(t-n)}{EMA(t-n)}.$$

The *Hall* criterion is simply the difference between the rolling maximum and the rolling minimum divided by the rolling average over n steps:

$$\mathcal{M}_{Hall} = \frac{\max(\mathcal{L}_{det}^n) - \min(\mathcal{L}_{det}^n)}{\text{mean}(\mathcal{L}_{det}^n)},$$

where $\mathcal{L}_{det}^n$ denotes the last $n$ values of $\mathcal{L}_{det}$.

The *Trend* metric is the slope of a linear regression trained on loss values in a window of size $n$, computed at each step using the closed-form solution for linear regression. The main issue of this criterion is its longer evaluation time compared to others.

The dynamics of all these metrics are shown in Figure 11. The main challenge we face with these metrics is their unstable behavior, which prevents the transfer of hyperparameters between concepts.

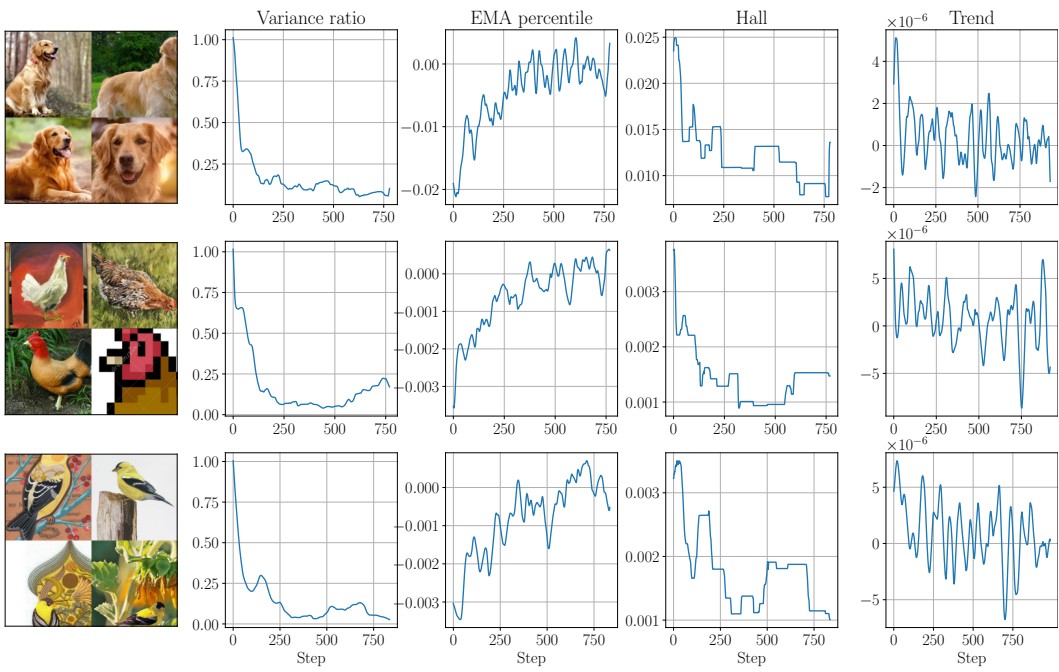

Figure 11: The dynamics of metrics for additional stopping criteria.

# F  Quantitative comparison with additional metrics

We conduct an additional quantitative evaluation of different early stopping approaches with two metrics introduced in [7] named DINO and DIV.

The DINO metric represents the average cosine similarity between ViT-S/16 DINO [29] embeddings of generated and real images. DINO is more sensitive to differences between subjects of the same class than CLIP image score, which is more suitable for measuring the degree of unique feature preservation. By contrast, DIV was proposed to evaluate the diversity of generated images. It is computed as the average LPIPS [30] cosine similarity between samples generated with the same prompt. Low values of DIV signify the model overfitting the training images.

Table 3: Comparison of identity preservation and diversity of speedup methods for three text-to-image customization techniques. Standard deviations over 48 concepts are in subscript.

| Metric | Baseline | CLIP-s | Few iters (mean) | Few iters (max) | DVAR |
|---|---|---|---|---|---|
| **Textual Inversion** | | | | | |
| DINO | $0.635_{\pm 0.094}$ | $0.590_{\pm 0.110}$ | $0.559_{\pm 0.133}$ | $0.591_{\pm 0.103}$ | $0.566_{\pm 0.119}$ |
| DIV | $0.742_{\pm 0.078}$ | $0.769_{\pm 0.097}$ | $0.768_{\pm 0.099}$ | $0.757_{\pm 0.092}$ | $0.777_{\pm 0.091}$ |
| **DreamBooth-LoRA** | | | | | |
| DINO | $0.721_{\pm 0.103}$ | $0.709_{\pm 0.093}$ | $0.711_{\pm 0.111}$ | $0.704_{\pm 0.125}$ | $0.577_{\pm 0.206}$ |
| DIV | $0.565_{\pm 0.154}$ | $0.630_{\pm 0.100}$ | $0.632_{\pm 0.086}$ | $0.592_{\pm 0.127}$ | $0.585_{\pm 0.114}$ |
| **Custom Diffusion** | | | | | |
| DINO | $0.475_{\pm 0.139}$ | $0.471_{\pm 0.140}$ | $0.475_{\pm 0.143}$ | $0.488_{\pm 0.145}$ | $0.454_{\pm 0.136}$ |
| DIV | $0.753_{\pm 0.061}$ | $0.748_{\pm 0.063}$ | $0.753_{\pm 0.069}$ | $0.756_{\pm 0.068}$ | $0.740_{\pm 0.055}$ |

Table 3 shows that DVAR increases the DIV metric for two personalization techniques, which indicates reduced overfitting, whereas having only minor losses in subject fidelity as measured by DINO. Overall, we conclude that our approach does not cause any significant quality degradations while decreasing customization time in an adaptive manner.

# G  Variance across concepts

To additionally justify the need for an adaptive early stopping technique, we conduct two experiments. First, we analyze the variability of $\mathcal{L}_{det}$ across concepts. Second, we compare the distribution of stop iterations selected by DVAR with those selected by non-adaptive early stopping methods.

## G.1  $\mathcal{L}_{det}$ variation over concepts

In Figure 12, we depict the behavior of normalized $\mathcal{L}_{det}$ for four concepts across all three personalization methods. Normalization is necessary, because the scale of loss varies highly across concepts.

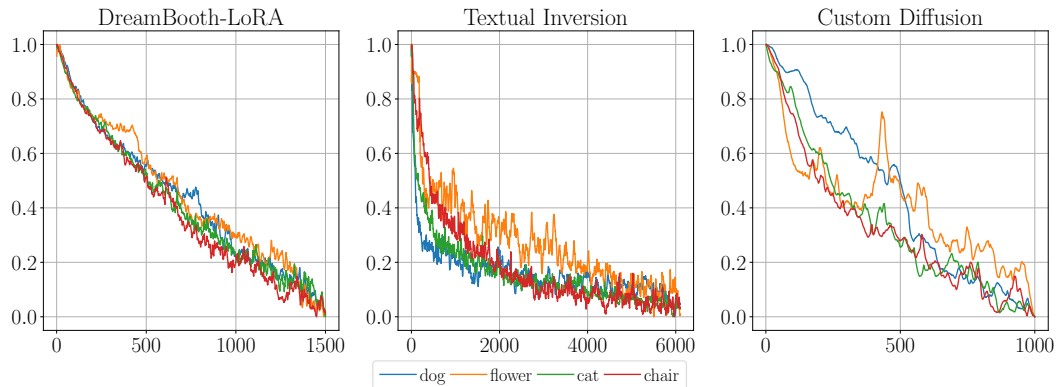

Figure 12: Behavior of $\mathcal{L}_{det}$ for various concepts and methods.

From this plot, we can conclude that the dynamics of $\mathcal{L}_{det}$ moderately differ from one concept to another. For example, the objective function exhibits earlier saturation on some concepts and methods and demonstrates a more unstable behavior on others.

## G.2  Final steps distribution

We compare the distribution of the final steps selected by DVAR to the non-adaptive methods. As we can see from Figure 13, simply throwing away some of the iterations might be sub-optimal for some concepts, indicating the need for an adaptive early stopping technique.

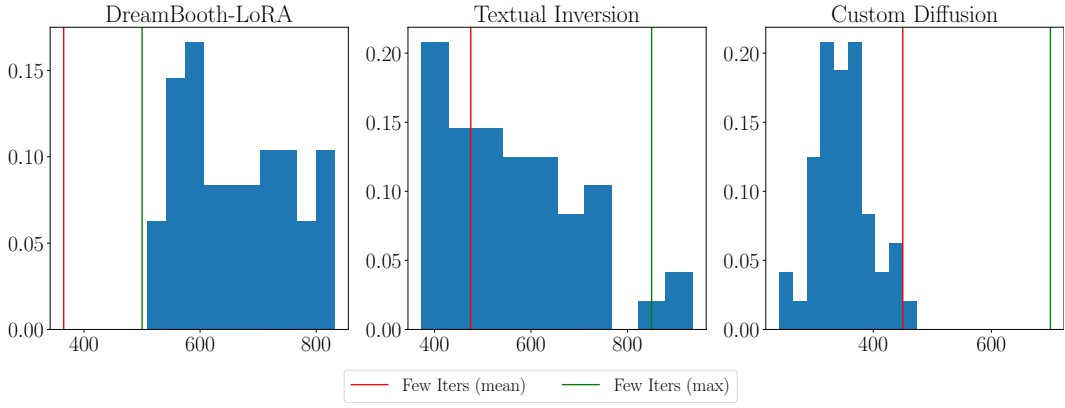

Figure 13: DVAR stop iterations distribution across concepts in relation to non-adaptive baseline.

## H  Qualitative comparison

In Figure 14, we show a side-by-side comparison of DVAR and Baseline methods. Validation images are obtained from the final training checkpoints using the validation prompt. This comparison confirms that our method achieves similar results in terms of concept reconstruction and prevents overfitting for all three adaptation approaches.

Moreover, this comparison demonstrates that focusing only on the quality of samples during training can be detrimental to the model's ability to transfer concepts to unseen contexts. In such cases, the model tends to generate outputs that resemble the training images regardless of the input prompt.

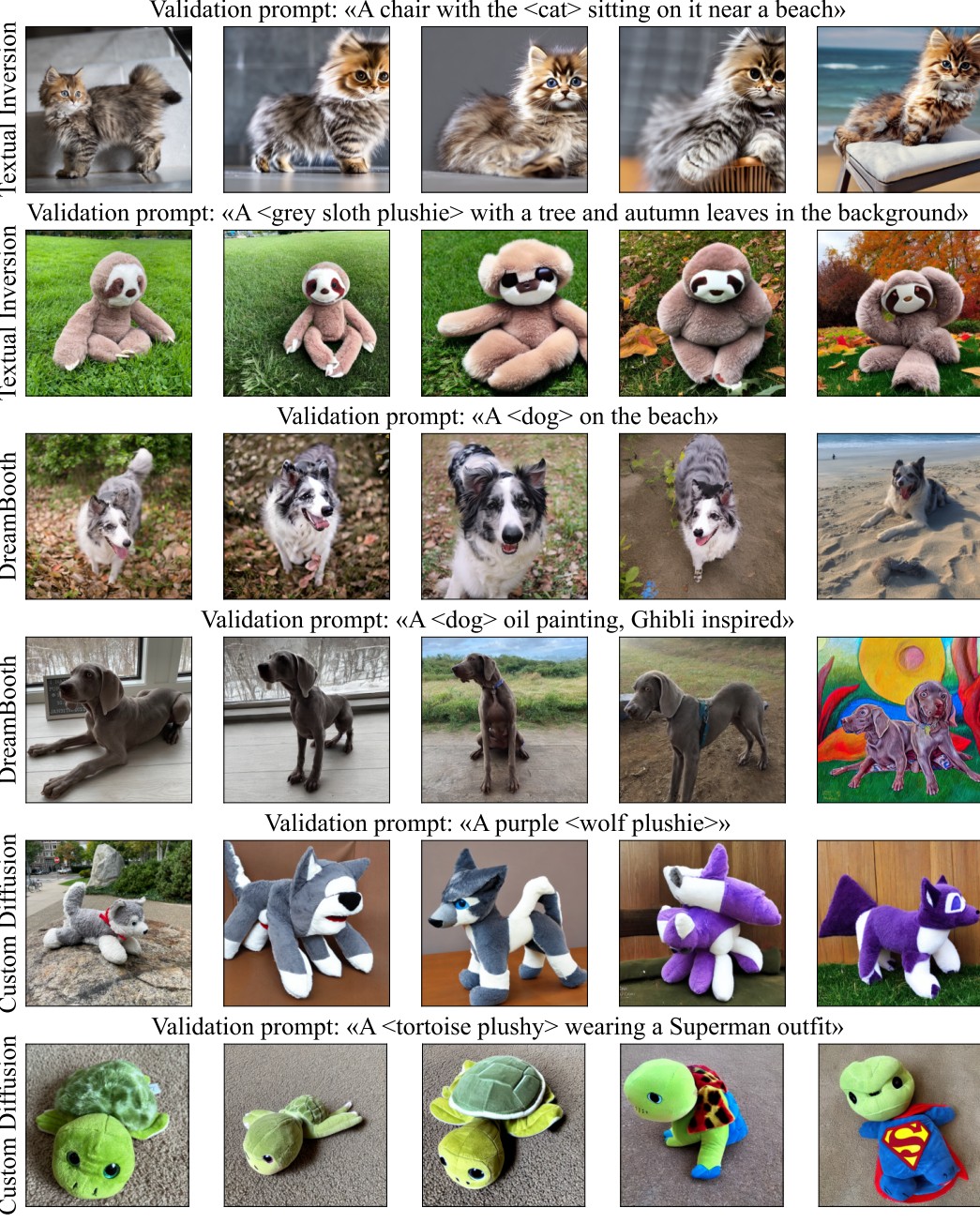

Figure 14: Side-by-side comparison of DVAR reconstruction quality and customization ability in relation to the baseline.

# I Human evaluation instructions

In Figure 15, we provide the instructions for the quality assessment tasks that were shown to the annotators. We aimed to make the instructions as unambiguous and detailed as possible, minimizing the potential influence of different interpretations among individuals. All annotators were required to complete a training task specifically designed to ensure a clear understanding of the decision-making algorithm. Furthermore, we incorporated control tasks with obvious correct answers to filter out bots or low-quality annotators who might vote randomly.

**Task description:**
Two neural networks tried to learn the object depicted by the Reference image and to generate it in another environment (e.g. another background, view angle, style or composition) described in text prompt on the top of the task. Please help us understand which neural network does the job better in terms of matching the text prompt by answering one question for a given pair of generated images.

**How to answer the question:**

1. Carefully look at the reference image. First of all, try to coarsely describe the class of the given image (e.g. table, dog, toy, backpack, etc.). Then pay attention to details, try to figure out what distinguishes this specific object from another from its class: its shape, texture, coloring, additional elements or specific details, etc.

2. If the reference object is present only on one of the pictures select the picture where it is present. If the reference object is not present on any of the pictures select Equal. Else: compare two generated images by their resemblance to the reference. We suggest going from the most general traits to the more specific details:

3. Which of the generated images better captures the class of the original object? If both images equally reflect the coarse class of the object, proceed to step 4. Otherwise, please select a corresponding image.

4. Which of the generated images better reflects the shape of the object being learned? If you think that both images equally reflect the shape of the object, proceed to step 5. Otherwise, please select a corresponding image.

5. Which of the generated images better reflects the colouring of the object being learned? If you think that both images reflect the object's color equally, proceed to step 6. Otherwise, please select a corresponding image.

6. Are there any details that one generated image reflected better than another? If yes, please select a corresponding image. Otherwise, please select "Equally similar".

| Reference: | Generated image A: | Generated image B: |
|---|---|---|
| <REFERENCE IMAGE> | <GENERATED IMAGE> | <GENERATED IMAGE> |

**Which of the generated images resembles the reference more?**

(a) Instructions for Reconstruction survey.

**Task description:**
Two neural networks tried to learn the object depicted by the Reference image and to generate it in another environment (e.g. another background, view angle, style or composition) described in text prompt on the top of the task. Please help us understand which neural network does the job better in terms of matching the text prompt by answering one question for a given pair of generated images.

**How to answer the question:** Each prompt (text description that was used to generate image) contains a word or phrase in brackets like <cat-statue> (this is a unique identifier of the object being learned by the network) and the description of a scene in which we want to see this object depicted. Descriptions may include specifications of different places, circumstances, change of reference object color, texture, shape, etc.

1. First, read the prompt replacing the object identifier with the word "something". For example if the prompt is "A <cat-statue> in front of the Eiffel Tower", read the prompt as "A something in front of the Eiffel Tower".

2. Now try to identify which image better matches this modified prompt. If one image is better than another, please vote for the corresponding image. Else: proceed to the next step.

3. Now put back the reference object identifier into the prompt and try to answer the following question: "Which image have a <reference-image-object> incorporated into the scene?"

4. To do so, first carefully study the reference image object: try to coarsely classify it, pay attention to its shape, coloring, texture, any patterns and details.

5. Now, for example our example prompt "A <cat-statue> in front of the Eiffel Tower" if one image depicts a random object in front of the Eiffel Tower and the other depicts something similar to the <cat-statue> (e.g. a cat toy or a cat statuette of the similar shape and/or coloring. More specific details are not crucial), the latter image should be chosen.

6. If one image incorporates learned object into the scene and another does not or does it in some details better than another, please select corresponding image. Else: please vote "equal".

**Text description:** <Text description>

| Reference: | Generated image A: | Generated image B: |
|---|---|---|
| <REFERENCE IMAGE> | <GENERATED IMAGE> | <GENERATED IMAGE> |

**Which of the generated images matches the text description more?**

(b) Instructions for Customization survey.

Figure 15: Human annotation interface.

## J  Correlations of timesteps sampling intervals and metrics

To investigate the impact of varied timesteps, we divide the interval from $0$ to $1000$ into three equal parts (beginning, middle, and end) and evaluate the $\mathcal{L}_{det}$ with timesteps sampled from each subinterval. Next, we run the training process over $1500$ steps for 9 concepts, with sampling images every 50 steps and calculating four CLIP-based quality metrics: train image similarity, train text similarity, validation image similarity, and validation text similarity.

Table 4: Pearson correlations between losses and quality metrics for different timestep sampling intervals. All results are statistically significant (p-value $< 0.01$)

|  | Begin (0-333) | Middle (333-666) | End (666-1000) | Full (0-1000) |
|---|---|---|---|---|
| Train text score | 0.27 | 0.33 | 0.30 | 0.36 |
| Validation text score | 0.48 | 0.51 | 0.37 | 0.50 |
| Train image score | -0.63 | -0.64 | -0.56 | -0.62 |
| Validation image score | -0.58 | -0.60 | -0.53 | -0.56 |

Inspired by previous research [15], we hypothesize that different timestep intervals might exhibit similarities with different metrics. However, as indicated in Table 4, our results do not reveal such behavior. The values within the rows demonstrate remarkable similarity to each other across all subintervals, even though there are different trends for particular metrics.

