# OpenReview forum: "Is This Loss Informative? Faster Text-to-Image Customization by Tracking Objective Dynamics"
_NeurIPS.cc/2023/Conference — NeurIPS 2023 poster_

### Official Review · Reviewer_rpQb · 2023-07-04

**Soundness:** 3 good
**Presentation:** 2 fair
**Contribution:** 3 good
**Rating:** 6
**Confidence:** 3

**Summary:**

Text-to-image generation models offer fine-grained control over synthesized images, but fast adaptation to smaller datasets or new concepts remains a challenge. Existing efficient adaptation methods suffer from long training times, hindering practical applications and resource usage. This work addresses the issue by studying the training dynamics of popular text-to-image personalization methods and proposes a drop-in early stopping criterion that significantly speeds up adaptation (up to 8 times faster) without compromising quality, as demonstrated through experiments on Stable Diffusion and various personalization methods.

**Strengths:**

1. This paper proposes a simple but effective method to accelerate text-to-image customization.
2. The proposed method is well-motivated and easy to understand.

**Weaknesses:**

1. It is imperative to provide supporting evidence to justify the necessity of adaptive step choices. Can we simply set a fixed step number (e.g. reduce to 1/3) without losing much performance? For instance, analyzing the outcomes and plotting the distribution of selected step numbers can demonstrate the potential reduction in unnecessary iterations. This approach would enhance the validity of the proposed method.
2. The authors should engage in a thorough discussion of pertinent literature concerning the acceleration of generative models. It is crucial to acknowledge and reference closely related research in this domain.

---

Having read the author's rebuttal, I've chosen not to alter my score.

**Questions:**

Please address the issues highlighted in the Weaknesses section.

**Limitations:**

Please address the issues highlighted in the Weaknesses section.

---

> ### Author Rebuttal · Authors · 2023-08-09
>
> Thank you for your valuable feedback and insightful suggestions! Allow us to address your concerns in a below response:
> >It is imperative to provide supporting evidence to justify the necessity of adaptive step choices. For instance, analyzing the outcomes and plotting the distribution of selected step numbers can demonstrate the potential reduction in unnecessary iterations.
>
> We agree that the justification of adaptive number of steps is highly required. This is why in **L262-265** we mention that non-adaptive Few Iters baseline has a **greater variance of almost all reported metrics**, which indicates a need for an adaptive early stopping method. However, plotting the distribution of selected steps might indeed be even more illustrative. Thank you for this suggestion: we include such a plot in the PDF of the general response.
> >The authors should engage in a thorough discussion of pertinent literature concerning the acceleration of generative models.
>
> Acceleration of generative models is indeed closely related to our work’s area of research. However, since DVAR does not change neither the optimization objective nor the sampling procedure, **all advancements in this area are complementary to our research**. In other words, one can use any of such techniques to accelerate the training process while using $L_{det}$ to track the convergence of the model. Keeping our paper succinct yet thorough required us to be selective about the related works that we cover. Nevertheless, we agree with your suggestion, and we are happy to discuss ways of accelerating generative models in an additional paragraph of Appendix A. We would be especially grateful for your suggestions of particular research areas that we should cover.

---

### Official Review · Reviewer_Gk9z · 2023-07-09

**Soundness:** 2 fair
**Presentation:** 2 fair
**Contribution:** 2 fair
**Rating:** 5
**Confidence:** 4

**Summary:**

This paper studies the training dynamics of a few state-of-the-art txt-to-image personalization methods and proposes an early stopping criteria while fine-tuning the base models to speedup their customization. They evaluate this criteria on Dreambooth, Textual inversion, and custom diffusion on 18 concepts and showing close or slightly better img-txt clip similarity scores on the validation prompts while degrading the similarity to the input image.

**Strengths:**

1. This paper presents a comprehensive study on the randomness factors during training of the customization models and their training curves and propose a simple stopping criteria based on a deterministic variance evaluation of the diffusion loss.
 2. This early stopping results in a 2x speedup on custom diffusion as well as dreambooth-LoRA and 12x speedup in textual inversion.

**Weaknesses:**

1. The whole idea of the paper is not very novel, and looks more like an analysis paper.
1. There is a trade-off in the samples similarity to the source image during both training and validation versus the number of finetuning steps. Although the authors have shown more generalization ability to validation prompts, it looks like identity preservation has been degraded.
2. The evaluations are only done on 18 concept examples. A larger set would be needed to make the results more convincing.
3. The scores which are reported in Table 2, only measure the similarity of the generated image with the validation text prompt. It would be important to see how much of the identity is preserved in these generations using the proposed early stopping criteria. Identity preservation is an important goal in personalization methods and is not studied well in this paper.

**Questions:**

1. In addition to the above questions mentioned above, it was not clear to me if the models are trained on all concepts together or if they are trained on the samples of each concept. If it is on each concept, where are 8 images per batch coming from?
If that has been done on multi-concepts, then I'd expect to see some evaluations on multi-concept customization.

2. Very few qualitative examples are shown in the paper/supplemental. It would be interesting to add more examples comparing the results with the baseline.

**Limitations:**

- The proposed technique causes a natural trade-off between the number of fine-tuning iterations and identity preservation of the objects in the source images. While the latter is quite important in image personalization, it is degraded in this paper (based on Table 1) and is not studied well in human evaluations.

---

> ### Author Rebuttal · Authors · 2023-08-09
>
> Thank you for taking the time to provide detailed feedback on our work. Please find our responses to your concerns and questions below:
> >The whole idea of the paper is not very novel, and looks more like an analysis paper.
>
> To the best of our knowledge, **we are the first to to study the optimization process of text-to-image customization** in detail and to propose a method to **make the loss of diffusion models informative**. If there are **specific studies** that you can direct us to where such an analysis has been conducted and a solution has been proposed, we would be grateful to learn about them.
>
> Moreover, we politely disagree that positioning our work as an analysis paper should necessarily be viewed as its weakness. We believe that providing comprehensive analysis presents its own set of equally important contributions for the scientific community.
> >There is a trade-off in the samples similarity to the source image during both training and validation versus the number of finetuning steps.
>
> You are correct: **we explicitly indicate in L269-270** that this trade-off exists and is one of the problems of the baseline methods we employ in our paper. However, **this is not a disadvantage of our method**. On the contrary, Table 2 reveals that although the baseline method often outperforms DVAR in terms of the reconstruction quality, it considerably lags behind our method in terms of the **customization** ability. Hence, our method allows us to identify an **optimal iteration to stop training** from the perspective of this trade-off. As evidenced in Table 1, early stopping by DVAR results in a negligible decrease in the quality of reconstruction on average across all concepts while preserving high quality of customization.
> >The evaluations are only done on 18 concept examples. A larger set would be needed to make the results more convincing.
>
> We agree and conduct additional experiments on **30 new concepts** from the DreamBooth paper. Please see the second paragraph of our general response for the updated results.
> >The scores which are reported in Table 2, only measure the similarity of the generated image with the validation text prompt. It would be important to see how much of the identity is preserved in these generations using the proposed early stopping criteria. Identity preservation is an important goal in personalization methods and is not studied well in this paper.
>
> We kindly disagree, because identity preservation is studied throughout the work. First, identity preservation is quantitatively evaluated by the **Val CLIP img** metric in Table 1. Similarly to the explanation of the Train CLIP img score provided in L229-231, Val CLIP img score measures identity preservation on **images generated from novel prompts**: we will emphasize that in the setup description to avoid reader confusion in the future. Judging by this metric, we see that DVAR is comparable to the baseline on DreamBooth-LoRA, slightly surpasses the baseline on Custom Diffusion, and slightly underperforms on Textual Inversion. Importantly, the possibility of identity preservation depends on the method and not just on the early stopping technique: training for more iterations leads to better identity preservation, but also to overfitting.
>
> Second, in Appendix F, Figure 13b provides instructions according to which the annotators determine the degree of customization. The third point of the instruction asks annotators to answer the question "Which image has a **<reference-image-object> incorporated into the scene?**" and the fifth point asks to determine **which image preserves the identity better**. Therefore, **identity preservation is covered by the Customization metric in Table 2** as well.
> >it was not clear to me if the models are trained on all concepts together or if they are trained on the samples of each concept. If it is on each concept, where are 8 images per batch coming from?
>
> The models were trained on the samples of each concept separately; we leave multi-concept personalization out of the scope of our work. Following Equations (2) and (3) and L100-102, the training  batches consist of images (x ~ X, where X is the 3 – 5 images for each concept) with minimal augmentations (central/horizontal crop), randomly sampled captions (y ~ Y), timesteps (t ~ U[0, T]), and random noise from the multivariate Gaussian distribution. Therefore, each batch is generated from **a small set of original images with different random inputs**: note that DVAR does not fix them for the training objective, only for evaluation. Thanks for highlighting this point of confusion, we will clarify the training process in the camera-ready version.
> >Very few qualitative examples are shown in the paper/supplemental.
>
> Besides the side-by-side comparisons in Figures 4 and 13 of our paper, we provide additional qualitative comparison in the PDF attached to our general response.

---

> > ### Comment · Reviewer_Gk9z · 2023-08-20
> > **Revised rating**
> >
> > Thanks for the rebuttal. Since most of my concerns are addressed in this rebuttal, I'd be happy to increase my rating.

---

### Official Review · Reviewer_DXLz · 2023-07-10

**Soundness:** 4 excellent
**Presentation:** 3 good
**Contribution:** 3 good
**Rating:** 7
**Confidence:** 5

**Summary:**

This paper argues that customization techniques for diffusion-based text-to-image generation models train for longer than is needed. This is because the training loss of diffusion models is often not informative -- i.e., often looks like stationary noise -- so practitioners tend to use a fixed (often excessive) number of training steps. This paper identifies and analyzes the sources of stochasticity in the training loss, and propose simple ways to eliminate them to make the loss more informative. They also introduce a simple early stopping criteria based on this interpretable loss.

**Strengths:**

Paper addresses a key issue many people training diffusion models face: the loss is not informative. That is, it often behaves like stationary noise despite the model continuing to improve on auxiliary metrics of interest (FID, human evals).

The authors do a very principled analysis of the sources of stochasticity in this loss and identify the sampled time-step to be the main driver. Eliminating this randomness leads to loss that better corresponds to model performance. I could se this becoming common practice -- not just in the model customization/fine-tuning regime but also in the training of the base diffusion model.

**Weaknesses:**

The main weakness of the paper is that I'm not sure how useful the DVAR early stopping criteria is. See question below.

However for this paper, this is less of an issue, since it's main contribution is a careful analysis of a problem people who train diffusion models face: uninformative loss and where it comes from.


**Questions:**

How does L_det vary across concepts/models/methods?

One plot that could help motivate the DVAR is a plot that shows how L_det varies across concepts (and models/methods). If it's the case that L_det doesn't vary very much then maybe picking a fixed (but smaller) number of fine-tuning steps is sufficient.

---

> ### Author Rebuttal · Authors · 2023-08-09
>
> Thank you for your review! We would like to address your concern in the following response:
> >How does L_det vary across concepts/models/methods?
>
> Different behavior of $L_{det}$ for different methods can be observed in Figures 2, 10 – 12 of our work. In Figure 2 of the PDF attached to our general response, we depict the behavior of normalized $L_{det}$ for 4 concepts across all three personalization methods. Normalization was necessary, because the scale of loss varies highly across concepts. From this plot, we can conclude that the dynamics of  $L_{det}$ moderately differ from one concept to another. For example, the objective function exhibits earlier saturation on some concepts and methods and demonstrates a more unstable behavior on others. We will include this illustration in the next revision of our paper — thank you for this suggestion!

---

### Official Review · Reviewer_5cki · 2023-07-11

**Soundness:** 3 good
**Presentation:** 2 fair
**Contribution:** 2 fair
**Rating:** 4
**Confidence:** 5

**Summary:**

This paper studies the training dynamics of popular text-to-image personalization methods (such as Textual Inversion, DreamBooth, and Costom Diffsuion), aiming to speed them up by an early stopping approach which allows the model to optimize or fine-tune for fewer iterations. A key observation is that most concepts are learned at early stages and do not improve in quality later, but standard model convergence metrics fail to indicate that. Based on this observation, the authors propose a simple drop-in early stopping criterion that only requires computing the regular training objective on a fixed set of inputs for all training iterations. Experiments are conducted on Textual Inversion, DreamBooth, and Costom Diffusion.

**Strengths:**

1. The observation that most concepts are learned at early stages and do not improve in quality later, but standard model convergence metrics fail to indicate that is interesting and inspiring.
2. The proposed approach is well-motivated with key observations and in-depth analysis before deriving the method.
3. The proposed approach improves the efficiency of personalized text-to-image models by a simple but effective early stopping scheme.

**Weaknesses:**

1. Although the visual results and CLIP scores indicate no further improvement after optimization for a certain number of steps, it is still not clear whether the observation is solid. The CLIP score can be biased because of the limited ability of CLIP in understanding complex and detailed information. Good visual results in generating an image similar to the input image do not mean perfect identity and detail perservation for personalized text-to-image generation.
2. The writing and logic can be improved. For example, how does 3.2 (investigating the sources of randomness) relate to other sections is not well demonstrated.
3. The evaluation is limited. Firstly, only 18 concepts are used for evaluation. Prior work such as DreamBooth actually used more concepts and prompts. Secondly, the CLIP image-image and CLIP image-text similarities are not enough to evaluate the concept preservation and image quality of personalized diffusion models. Prior work DreamBooth used several other evaluation metrics to reflect the ability of the models. Thirdly, the visual results in Figure 4 are not impressive, and there are no examples for personalized generation where the user provides a different text prompt with the same visual concept from the original image but in a different background or scene. The different application scenarios of DreamBooth, Textual Inversion, and Custom Diffusion are not extensively experimented.

**Questions:**

Please refer to the weakness section.

**Limitations:**

The authors addressed the limitations.

---

> ### Author Rebuttal · Authors · 2023-08-09
>
> Thank you for a detailed review! We would like to answer your concerns about our work below:
>
> >Good visual results in generating an image similar to the input image do not mean perfect identity and detail perservation for personalized text-to-image generation.
>
> We agree that the CLIP image score may not fully reflect the process of the model learning complex details. However, as indicated in Table 2 of the paper, we also conducted **human evaluation** to compare our method with the baseline in two directions: concept **reconstruction** and **customization**. Our results demonstrate that 1) our method often stops at an iteration **that is better or equal in terms of reconstruction** quality 2) this iteration also proves to be **the best for image customization**, indicating the overfitting of original methods which can be overcome by DVAR. Additionally, the correlation between the CLIP image score and the visual quality of reconstruction can be partially confirmed using Figures 2, 10 – 12.
>
> >The writing and logic can be improved. For example, how does 3.2 (investigating the sources of randomness) relate to other sections is not well demonstrated.
>
> Section 3.2 is necessary to **motivate and introduce the new objective** $L_{det}$ (a key factor in our work) and to explain how it differs from the original training objective. $L_{det}$ is used for the DVAR early stopping criterion, which is one of the primary contributions of this paper.
>
> >The evaluation is limited. Firstly, only 18 concepts are used for evaluation.
>
> We have conducted additional experiments on the DreamBooth dataset with **30 new concepts**; see the general response for details.
> Prior work DreamBooth used several other evaluation metrics to reflect the ability of the models.
> Thank you for this suggestion! We have updated the table with main results from our paper with **two more metrics** used in the DreamBooth paper (DINO and DIV). Please refer to the table with updated results in our general response.
> >there are no examples for personalized generation where the user provides a different text prompt with the same visual concept from the original image but in a different background or scene.
>
> Thank you for pointing that out. Actually, this setting is shown in Figure 4 of the submission, but it might be difficult to interpret because we did not specify the validation prompt. For example, in the row “Textual Inversion val”, the prompt “A photo of <clock> *on the beach*” is used to check if the learned concept can be used in a different scene. Both the baseline and CLIP-s methods **fail to depict** the learned concept in the desired background, whereas **DVAR succeeds**. Other qualitative examples are provided in Figure 13 of Appendix J. An improved version of Figure 4 and other qualitative side-by-side comparisons can be found in the PDF attached to our general response.

---

> > ### Comment · Reviewer_5cki · 2023-08-15
> > **revised rating**
> >
> > Thank the authors for the response. The authors have addressed my concerns about the limited concepts and metrics used for evaluation by adding more concepts and evaluation metrics in the rebuttal. The authors also explained that they showed the personalized generation results in the paper although it was not illustrated properly (prompts are not shown so it was difficult to understand what the images mean). This can be improved in the final version of the paper. My remaining concern is that the proposed approach does not seem to demonstrate good identity and detail preservation as shown in the examples in the paper and the additional rebuttal page. Therefore, I increase my rating to borderline reject.

---

### Author Rebuttal · Authors · 2023-08-09

Dear reviewers, we deeply appreciate the time and effort you devoted to the review of our paper. We are glad that multiple reviewers recognize the motivation that drives our work (**5cki**, **DXLz**, **rpQb**), the depth of our analysis (Gk9z, DXLz), and the simplicity of DVAR combined with its effectiveness (**5cki**, **rpQb**). We have carefully considered your insightful feedback and have addressed the concerns raised by each reviewer in their respective responses. In this response we would like to address your collective questions and comments.

First, we would like to address a common concern regarding the evaluation of the studied methods on only 18 concepts. In the submitted version of our work, we used all the concepts and prompts provided by the authors of two methods that we test: Custom Diffusion and Textual Inversion. The table below presents an extended evaluation on additional 30 concepts that were recently released by the DreamBooth paper authors (resulting in **48 concepts overall**).

The evaluation is additionally augmented with two more metrics from the latest revision of DreamBooth paper (following the suggestion from Reviewer **5cki**): DINO measures the preservation of subject details in the images generated from train prompts and DIV measures the diversity of the samples generated from the same prompt.

|              |    |  |    Textual Inversion       ||            |        |     |
|--------------------|-----------------|---------------|---------------|--------------|--------------|--------------|-----------
| Method              | Train CLIP img   | Val CLIP img   | Val CLIP txt   | DINO          | DIV           | Iterations       | Time, min    |
| Baseline             | $0.840_{±0.051}$    | $0.786_{±0.075}$  | $0.209_{±0.021}$  | $0.635_{±0.094}$ | $0.742_{±0.078}$ | $6100.0_{±0.0}$  | $27.0_{±0.3}$ |
| CLIP-s | $0.824_{±0.053}$    | $0.757_{±0.067}$  | $0.233_{±0.024}$  | $0.590_{±0.110}$ | $0.769_{±0.097}$ | $666.7_{±174.5}$ | $9.6_{±2.5}$  |
| Few Iters (mean)    | $0.796_{±0.069}$    | $0.744_{±0.073}$  | $0.232_{±0.023}$  | $0.559_{±0.133}$ | $0.768_{±0.099}$ | $475.0_{±0.0}$   | $1.6_{±0.0}$  |
| Few Iters (max)     | $0.806_{±0.066}$    | $0.767_{±0.071}$  | $0.219_{±0.022}$  | $0.591_{±0.103}$ | $0.757_{±0.092}$ | $850.0_{±0.0}$   | $2.8_{±0.0}$  |
| DVAR | $0.795_{±0.067}$    | $0.748_{±0.068}$  | $0.227_{±0.024}$  | $0.566_{±0.119}$ | $0.777_{±0.091}$ | $566.0_{±141.5}$ | $3.1_{±0.8}$  |
|              |    |  |    **DreamBooth-LoRA**       ||            |        |     |
Baseline             | $0.857_{±0.061}$    | $0.824_{ ±0.079}$  | $0.205_{±0.021}$  | $0.721_{±0.103}$ | $0.565_{±0.154}$ | $1000.0_{±0.0}$ | $8.1_{±2.2}$|
| CLIP-s | $0.862_{±0.045}$    | $0.788_{±0.075}$  | $0.225_{±0.022}$  | $0.709_{ ±0.093}$ | $0.630_{±0.100}$ | $353.2_{ ±88.1}$ | $6.1_{ ±1.5}$ |
| Few Iters (mean)    | $0.855_{ ±0.052}$    | $0.806_{ ±0.085}$  | $0.219_{±0.023}$  | $0.711_{±0.111}$ | $0.632_{±0.086}$ | $367.0_{±0.0}$  | $1.9_{ ±0.0}$ |
| Few Iters (max)     | $0.851_{ ±0.053}$    | $0.800_{ ±0.097}$  | $0.214_{ ±0.019}$  | $0.704_{ ±0.125}$ | $0.592_{±0.127}$ | $500.0_{ ±0.0}$  | $2.6_{ ±0.1}$ |
| DVAR | $0.784_{ ±0.106}$    | $0.687_{ ±0.140}$  | $0.238_{ ±0.034}$  | $0.577_{ ±0.206}$ | $0.585_{ ±0.114}$ | $665.3_{ ±94.9}$ | $4.9_{ ±0.7}$ |
|              |    |  |    **Custom Diffusion**       ||            |        |     |
Baseline             | $0.755_{ ±0.077}$    | $0.695_{ ±0.069}$  | $0.258_{ ±0.021}$  | $0.475_{ ±0.139}$ | $0.753_{ ±0.061}$ | $500.4_{ ±0.5}$   | $6.5_{ ±0.9}$ |
| CLIP-s | $0.757_{ ±0.076}$    | $0.691_{ ±0.069}$  | $0.258_{ ±0.023}$  | $0.471_{ ±0.140}$ | $0.748_{ ±0.063}$ | $510.4_{ ±134.1}$ | $9.7_{ ±3.0}$ |
| Few Iters (mean)    | $0.751_{ ±0.078}$    | $0.691_{ ±0.070}$  | $0.259_{ ±0.023}$  | $0.475_{ ±0.143}$ | $0.753_{ ±0.069}$ | $450.0_{ ±0.0}$   | $3.4_{ ±0.9}$ |
| Few Iters (max)     | $0.754_{ ±0.078}$    | $0.691_{ ±0.073}$  | $0.257_{ ±0.022}$  | $0.488_{ ±0.145}$ | $0.756_{ ±0.068}$ | $700.0_{ ±0.0}$   | $5.3_{ ±1.4}$ |
| DVAR | $0.742_{ ±0.074}$    | $0.693_{ ±0.066}$  | $0.259_{ ±0.022}$  | $0.454_{ ±0.136}$ | $0.740_{ ±0.055}$ | $348.1_{ ±46.6}$  | $3.4_{ ±1.0}$ |

Noticeably, adding 30 novel concepts does not change the relative ranking of early stopping methods: DVAR still allows for early stopping **without sacrificing the reconstruction quality** (judged by Train/Val CLIP img and DINO metrics) for two out of three evaluated customization methods. Moreover, models trained with DVAR demonstrate **higher customization quality** than the baseline (demonstrated by Val CLIP txt metric), while being **adaptive** and **non relying on the costly intermediate sampling**. Additionally, DVAR increases the DIV metric for two personalization techniques, signifying **reduced overfitting**.

Moreover, we provide additional plots and illustrations in the attached PDF. Specifically, the attachment contains more qualitative side-by-side comparisons (**Gk9z**), a corrected version of Figure 4 with validation prompts specified (**5cki**), the distribution of final step numbers (**rpQb**), and the $L_{det}$ behavior on various concepts (**DXLz**).

We hope that a more extensive evaluation of our method addresses your concerns and that the additional illustrations further confirm our findings. If you have any additional questions, we would be happy to respond to them during discussion.

---

### Decision · Program_Chairs · 2023-09-21

**Decision:**

Accept (poster)

**Comment:**

This paper argues that customization techniques for diffusion models train for longer than is needed. The authors analyze the sources of stochasticity in the training loss, and propose simple ways to eliminate them to make the loss more informative.

The authors have done a nice job during rebuttal. After rebuttal, the paper received mixed scores of 4567. The reviewers who gave a score of 4 and 7 both gave a confidence score of 5 for their reviews. On one hand, this paper provides a careful analysis of a key problem people who train diffusion models face: uninformative loss and where it comes from. The authors do a principled analysis of the sources of stochasticity and provide simple fixes to make the loss more informative. The observations are interesting and inspiring.

On the other hand, reviewers have questioned about the limited concepts used to evaluation the methods (18 concepts in total). During the rebuttal, the authors have added new results on an additional set of 30 concepts, and added 2 additional evaluation metrics. Another concern shared by two reviewers is that the proposed approach does not seem to demonstrate good identity and detail preservation as shown in the examples in the paper and the additional rebuttal page. The authors have argued that identity preservation is studied throughout the work.

Overall, the AC finds this paper interesting and the rebuttal relatively convincing. On balance, the AC thinks that the merits of the paper slightly outweigh its flaws, and decided to recommend acceptance by the end. The authors are highly encouraged to add new results and discussions into the main paper based on reviewers' detailed comments.